# Immune Responses and Transcriptomic Analysis of *Nilaparvata lugens* against *Metarhizium anisopliae* YTTR Mediated by Rice Ragged Stunt Virus

**DOI:** 10.3390/plants12020345

**Published:** 2023-01-11

**Authors:** Xuewen Li, Bang Zhang, Jiaxing Zou, Qianqian Li, Jianli Liu, Shouping Cai, Komivi Senyo Akutse, Minsheng You, Sheng Lin

**Affiliations:** 1State Key Laboratory for Ecological Pest Control of Fujian and Taiwan Crops, Institute of Applied Ecology, Fujian Agriculture and Forestry University, Fuzhou 350002, China; 2Ministerial and Provincial Joint Innovation Centre for Safety Production of Cross-Strait Crops, Fujian Agriculture and Forestry University, Fuzhou 350002, China; 3Key Laboratory of Green Control of Insect Pests (Fujian Agriculture and Forestry University), Fujian Province University, Fuzhou 350002, China; 4Fujian Key Laboratory of Forest Cultivation and Forest Products Processing and Utilization, Fujian Academy of Forestry, Fuzhou 350002, China; 5International Centre of Insect Physiology and Ecology, Nairobi P.O. Box 30772-00100, Kenya

**Keywords:** brown planthopper, entomopathogenic fungi, rice virus, innate immunity, RNA-seq

## Abstract

Plant viruses and entomopathogenic fungi (EPF) can both elicit immune responses in insects. This study was designed to clarify whether plant viruses could affect the efficacy of EPF and explore the immune responses of brown planthopper (BPH), *Nilaparvata lugens,* in response to different pathogen infections. In this study, a strain of *Metarhizium anisopliae* YTTR with high pathogenicity against BPH was selected and explored whether rice ragged stunt virus (RRSV) could affect its lethality against BPH. RNA-seq was used to detect the inner responses of BPH in response to RRSV and *M. anisopliae* YTTR infection. Results showed that *M. anisopliae* YTTR has strong lethality against BPH (RRSV-carrying and RRSV-free). RRSV invasion did not affect the susceptibility of BPH against *M. anisopliae* YTTR at all concentrations. At 1 × 10^8^ spores/mL, *M. anisopliae* YTTR caused a cumulative mortality of 80% to BPH at 7 days post-treatment. The largest numbers of differentially expressed genes (DEGs) was obtained in BPH treated with the two pathogens than in other single pathogen treatment. In addition, KEGG enrichment analysis showed that the DEGs were mostly enriched in immune and physiological mechanisms-related pathways. Both RRSV and *M. anisopliae* YTTR could induce the expression changes of immune-related genes. However, most of the immune genes had varying expression patterns in different treatment. Our findings demonstrated that RRSV invasion did not have any significant effect on the pathogenicity of *M. anisopliae* YTTR, while the co-infection of *M. anisopliae* YTTR and RRSV induced more immune and physiological mechanisms -related genes’ responses. In addition, the presence of RRSV could render the interplay between BPH and *M. anisopliae* YTTR more intricate. These findings laid a basis for further elucidating the immune response mechanisms of RRSV-mediated BPH to *M. anisopliae* infection.

## 1. Introduction

The brown rice planthopper (BPH), *Nilaparvata lugens* (Stål) is one of the most destructive rice pests and widely spreads in rice paddies in Asia. Indeed, feeding on rice phloem with its genotype mouthparts, it’s also an important insect vector that transmits rice ragged stunt virus (RRSV) and rice grassy stunt virus (RGSV) [1,2]. The damage of BPH severely limits rice production, leading growers to the excessive use of chemical pesticides as the main means to control BPH. Long-term excessive and unreasonable use of chemical pesticides has resulted in BPH resistance to multiple insecticide groups [2,3,4]. Liao et al. [3] found that BPH had developed varying degrees of resistance to a broad range of frequently used insecticides such as thiamethoxam. It was reported that some of the pesticides have a promotion role in the fecundity and development of BPH, which will accelerate its outbreak of the pest in the field [3,5,6]. Moreover, other irrational agronomic practices, such as the irrational use of nitrogen fertilizer, made the control of BPH even more challenging [2,7]. 

Entomopathogenic fungi (EPF) are major pathogens causing diseases in insect populations, and there exist over 1000 different kinds of fungi with lethal effects to the pest. Compared to chemical control, biocontrol using EPF is not only sustainable but also eco-friendly, and could prevent the development of resistance in the pest populations. Thus the use of EPF has great potential for pest management [8,9,10,11]. Currently, several EPF including *Metarhizium anisopliae*, *Beauveria bassiana*, and *Lecanicillium lecanii* have been reported to have high mortality effects against BPH [12,13]. *M. anisopliae*, an important taxon of EPF [14,15], is widely applied for the control of BPH [15,16]. Jin et al. [17] tested the effects of different *M. anisopliae* strains against 3rd-instar nymphs of BPH and reported that the cumulative corrective mortality rates ranged between 6.5–64.2% at 9 days post-treatment with the concentration of 100 spores/mm^2^. In addition, *M. anisopliae* with the concentration of 1 × 10^8^ spores/mL was found to cause over 80% mortality to adult BPH at 9 days after treatment [12]. Tang et al. [16] also demonstrated that when using *M. anisopliae* CQMa421 against newly emerged adult BPH, at the concentrations of 1 × 10^5^ and 1 × 10^8^ spores/mL, the LT_50_ values were about 8 and 5 days respectively. However, the combination of *M. anisopliae* CQMa421 and insecticides produced a synergistic effect in suppression BPH. In addition to killing BPH through direct spraying (inundative application), *M. anisopliae* may also act as endophytes by colonizing a host plant to systemically control BPH [18,19]. 

Insects possess a strong immune system to defend themselves against external pathogens invasion, and it plays an indispensable role in promoting insects’ community prosperity [20,21]. Insect innate immunity contains cellular and humoral immunity, which work together to guard against alien pathogens [22,23,24,25,26]. The advances in omics techniques have led to an upsurge in the field of insect immunity [13,27,28,29,30,31,32]. In humoral immunity, when pathogens were recognized by pattern recognition receptors (PRRs), the relevant signaling pathways were triggered including Toll, Immune deficiency (Imd), Janus kinase/signal transducers and activators of transcription (JAK/STAT), Prophenoloxidase (PPO), and RNA interference (RNAi) pathways. The antimicrobial substances were then released to attack the harmful pathogens [22,33]. Plant viruses, like other foreign pathogens in the insect system, can trigger the immune response of their host insect vector [13,32]. Yoshikawa et al. [34] reported that both BPH-transmitted viruses (RRSV and RGSV) caused changes in the activity of detoxification enzymes, but at different degrees for different viruses, sexes, and resistant strains. Plant viruses can persist in vector insects, and they co-evolve in more intricate interactions with vector insects [33,35,36]. They can manipulate the host’s immune response by weighing their pathogenicity and proliferation, enabling long-term existence in the vectors [33,36]. Additionally, RRSV can cause apoptosis in the salivary gland cells of BPH, but only some regions were affected [37]. 

Plant viruses and EPF could co-exist in hosts where both have complex immune interactions with vector insects [22,33,35]. It is therefore important to understand whether RRSV invasion could affect the lethality of the entomopathogenic fungus *M. anisopliae* YTTR and the immune interactions between *M. anisopliae* YTTR and BPH. In addition, how BPH responds to infection by the two external pathogens is still unclear. Our previous study showed that *M. anisopliae* YTTR has a highly lethal effect on BPH (unpublished data). All tested individuals died at 12 days post-treatment at the concentration of 1 × 10^8^ spores/mL and with a LT_50_ value of 5 days. The body of BPH was stiff after infection with *M. anisopliae* YTTR, followed by hyphae and conidia development, which is consistent with the general symptoms developed after EPF infection of pests (Appendix A). In this study, we assessed whether RRSV infection could affect the virulence of *M. anisopliae* YTTR against BPH. The study further investigated the RRSV-mediated immune response of BPH to *M. anisopliae* YTTR infection and the response of BPH to the two different pathogens (*M. anisopliae* YTTR and RRSV) using the RNA-seq approach. 

## 2. Results

### 2.1. Effect of RRSV on the Lethality of Metarhizium anisopliae YTTR against BPH

To explore RRSV-mediated effects on the pathogenicity or virulence of EPF to BPH, *M. anisopliae* YTTR was used to treat the RRSV-carrying and RRSV-free adult females BPH. The results showed that the survival rate of adult females was decreased with an increased concentration of *M. anisopliae* YTTR and days after treatment, whether in the presence or absence of RRSV (Figure 1). Compared to RRSV-free BPH, RRSV invasion did not significantly (*p >* 0.05) affect the survival rate of BPH after *M. anisopliae* YTTR infection at all concentrations. 

### 2.2. Transcriptome Data Quality Control and Gene Annotation Analysis

RNA-seq was used to explore the effect of RRSV on the responses of BPH to *M. anisopliae* YTTR infection. RRSV-free BPH was treated with Tween-80 (CK) or *M. anisopliae* YTTR (SF), and RRSV-carrying BPH was also treated with Tween-80 (SV) or *M. anisopliae* YTTR (CI), respectively. After checking the quality of total RNA and sequencing, each sample generated an average of 49.2 G raw reads. The clean reads of each sample were above 6.8 GB, the GC content ranged between 46.69–48.33%, and the Q30 quality was higher than 92%. The percentage was uniquely mapped with the reference genome in 12 samples ranging from 65.81% to 68.97%, and the total matching rate was higher than 79.73% in all samples (Appendix A). All the data indicated the high quality of RNA-seq of each sample. 

All clean reads were annotated with the major databases (NR, Swiss-Prot, Pfam, EggNOG, GO, and KEGG). As shown in Appendix A, 11,753, 11,089, 16,080, 19,537, 13,355, and 14,947 genes were obtained from the six databases, respectively. Among them, the NR database has the largest number of annotated genes: 19,537. 

### 2.3. Differential Expressed Genes Analysis

To clarify the expression of DEGs among the four treatment groups (CK, SF, SV, and CI), every two or pairs treatment groups were compared based on the DEGs analysis where six comparison groups (CI vs. SF, CI vs. CK, CI vs. SV, SV vs. CK, SF vs. CK, and SF vs. SV) were defined. In each comparison group, 256, 338, 166, 258, 149, and 195 DEGs were obtained respectively (Figure 2 and Appendix A). Among these groups, 132, 147, 87, 110, 78, and 107 DEGs were up-regulated, respectively (Figure 2 and Appendix A). The greatest number of DEGs were obtained in CI vs. CK, which included 147 DEGs that were up-regulated and 191 DEGs that were down-regulated (Figure 2 and Appendix A). This indicates that co-infection with RRSV and *M. anisopliae* YTTR arouses more strong responses for BPHs genes’ expression. 

### 2.4. Venn’s Diagram Analysis

According to the result of Venn’s diagram analysis (Figure 3), 218 (37.91%) DEGs were solely expressed in CI, which expression may be attributed to RRSV and *M. anisopliae* YTTR invasion. These DEGs were mapped to 174 pathways in the KEGG database (Appendix A) and these pathways were involved in six categories, among which 11 pathways were annotated to the category of the immune system. In addition, some immune-related signal transductions such as the TNF signaling pathway and MAPK signaling pathway were found. Fifty (22 + 28) (8.70%) DEGs were co-expressed between CI vs. CK and SF vs. CK, which may be responsible for *M. anisopliae* YTTR invasion, while 98 (70 + 28) (15.65%) DEGs were expressed in the comparison groups of CI vs. CK and SV vs. CK, indicating that these genes were induced after RRSV invasion. Furthermore, 28 (4.87%) DEGs were co-expressed among the three comparisons (SF vs. CK, SV vs. CK, and CI vs. CK), which mapped to 14 pathways according to KEGG annotation. Most pathways were also performed with functions related to immunity (Appendix A). 

### 2.5. KEGG Enrichment Analysis 

The results of KEGG enrichment showed that KEGG was mainly enriched in immune, metabolic, and signal transduction-related pathways in the six comparison groups (Figure 4). The number of DEGs in significantly enriched pathways was much higher in CI vs. CK and CI vs. SF than in the other four comparison groups (padjust < 0.05) (Figure 4A,D). After pathogens invasion, most immune-related pathways such as Toll and Imd signaling pathway, and MAPK-related pathways were significantly enriched. Toll and Imd signaling pathway was found in all the six comparisons with enriched different levels, especially in the groups CI vs. CK, SF vs. CK, and SV vs. CK (Figure 4A−C) (padjust < 0.05). Indeed, immune-related pathways, pathways participating in vital physiological processes were also enriched. Amino acid (arginine and tryptophan) synthesis, nutrition digestion and absorption (protein, vitamin, and fat), and harmful substances (caffeine and alcoholism) metabolism-related pathways were also significantly enriched. The longevity regulating pathway was enriched in comparisons containing CI, which may be attributed to the co-infection of the two pathogens arousing more longevity-related genes response (Figure 4A,D,E). 

### 2.6. Immune-Related Genes’ Expression Level in Transcriptome Data

To further understand the immune-related genes’ expression in the different treatments, insect major humoral immunity pathways (PRRs, Toll pathway, IMD pathway, JAK-STAT pathway, Serine protease cascade, PPO cascade, and JNK pathway) were selected. The 66 immune-related genes were selected in transcriptome data and the results showed that different genes had different expressions among the four treatments (CK, SF, SV, and CI) (Figure 5; Appendix A). In PRRs, after pathogens treatment, the relative expression level of *galection-4* was enhanced significantly in SF, SV, and CI; and *PGRP-LB* and *dscam2* expression in the three treatments were also higher than that in CK. However, *β-1,3-GBP* expression was reduced greatly in pathogens infection treatments (SF, SV and CI) (Figure 5A; Appendix A). For the Toll pathway, most genes’ expression did not differ significantly among the four treatments. While the expression of *myd88* and *defensin* showed more extensive variations among the four treatment groups (CK, SF, SV, and CI). *Myd88* was down-regulated in SF and SV, while *defensin* was highly expressed in SF and CI (Figure 5B; Appendix A). In the IMD pathway, *IKK* and *IAP-1* were highly expressed in SV and CI, while, *IAP-1′*s expression in SF was significantly reduced compared with that in the other three treatments (CK, SV, and CI). After *M. anisopliae* YTTR treatments (SF and CI), the expression of *ankyrin3* was reduced compared with that in CK (Figure 5C; Appendix A). In addition, *Hopscotch*’s expression was reduced significantly after pathogens (*M. anisopliae* YTTR and RRSV) infection. Additionally, the expression of *SCS5* was enhanced in the three treatments (SF, SV, and CI) compared with that in CK (Figure 5D; Appendix A). For the cascade of Serine protease and PPO, the largest number of genes was annotated in these cascades and most genes’ expressions were down-regulated. The expression of *SP7* in SF was significantly higher than that in the other three treatments (Figure 5E; Appendix A). However, the highest expression of *TNF-α* was found in SF, and the lowest in CI (Figure 5F; Appendix A).

### 2.7. Transcriptomic Data Validation 

Ten DEGs were randomly selected in each comparison group to verify the reliability of the transcriptomic data using RT-qPCR. The results showed that the expressions of these selected genes in RT-qPCR were consistent with the transcriptome results, indicating the reliability of the transcriptome data (Figure 6, Appendix A). 

The four genes (*Toll*, *Myd88*, *PPO,* and *TNF-α*) in Appendix A were also found in the transcriptome data (Figure 5). According to the comparison results of the four genes’ relative expression levels in RNA-seq and RT-qPCR (Appendix A), 75% of the comparison results were consistent. Additionally, all the gene expressions differences in the pair of CI vs. CK were consistent between transcriptome data and RNA-seq data. 

## 3. Discussion

*M. anisopliae* is an important species of EPF, which is widely used for the management of BPH [12,15,16,18]. In our study, it was found that the *M. anisopliae* YTTR had high pathogenicity to BPH. Our findings showed that, when the concentration of *M. anisopliae* YTTR reached 1 × 10^8^ spores/mL, all the infected individuals can be killed within 14 days after treatment. The high pathogenicity of the fungus shows a strong potential of *M. anisopliae* YTTR in controlling the pest under field conditions. In rice paddies, the moist or humidity and hidden environment could significantly facilitate the growth of *M. anisopliae* YTTR and could consequently reduce its degradation under UV in the rice cropping systems. 

Plant viruses have a complex mutualistic relationship with their vector insects. They can persist within their vectors for a long time and achieve long-lasting transmission by regulating the interactions between their pathogenicity and proliferation [33,35]. Most studies of immune interactions between rice viruses and vector insects are mainly focused on *Laodelphax striatellus* and its transmitted rice viruses, such as the rice stripe virus (RSV) [36,38,39]. Zhou et al. [40] showed that the chronic presence of RSV in *L. striatellus* could suppress the expression of antiviral genes, *TLR13,* and promoting virus transmission. There also exists a complex interplay between RRSV and BPH [34,37]. In this study, we demonstrated whether the effect of RRSV on the immunity of BPH could produce any supplement impact on the efficacy of *M. anisopliae* YTTR. Our findings showed that RRSV did not affect the susceptibility of BPH against *M. anisopliae* YTTR at all concentrations, indicating that RRSV infection had no significant effect on the virulence of *M. anisopliae* YTTR against the insect pest. This result was consistent with the findings reported by Garza-Hernández et al. [41], who reported that the Dengue virus did not affect the *M. anisopliae* fatal efficacy against *Aedes aegypti* (L.) (Diptera: Culicidae). To further explore RRSV-mediated immune responses of BPH to the fungal infection, *M. anisopliae* YTTR was used for RNA-seq at a concentration of 1 × 10^8^ spores/mL. 

The advances in omics technology have provided a convenient vehicle for studying the interactions between insects and pathogens, where the use of RNA-seq to study insect innate immunity has become more prominent [13,27,32]. Based on our previous study and other related studies [13], after 12 h post-treatment with Tween-80 or *M. anisopliae* YTTR, the four treatments (CK, SF, SV, and CI) were prepared to proceed with transcriptome sequencing. Our results showed that the largest numbers of DEGs were obtained in CI vs. CK, with 147 up-regulated genes and 191 down-regulated genes. This indicates that BPH treated simultaneously with the two pathogens expressed more DEGs than any single pathogen treatment. Fifty (8.70%) DEGs were co-expressed between the group of SV vs. CK and SF vs. CK, and these genes may play a common anti-pathogen role against fungi or viruses. Compared with CK, SF, SV, and CI treatments down-regulated the expressions of *PGRP-LF*, *β-1,3-GBP*, *Ankyrin3*, *Hopscotch*, *SPI*, *SP-snake*, *SPIB5*, *SPIB10*, *Lyzome2*, *I-Lysozyme3*, and *TNF-α-H*, which may be attributed to the negative relationship with the pathogens (*M. anisopliae* YTTR and RRSV), and up-regulated the expression of *PGRP-LB*, *Galectin-4*, *Dscam2*, *Defensin*, *SCS5,* and *PPAF* [36,40]. Some genes were only highly expressed in one pathogen treatment, which may be due to the fact that these genes have a special anti-pathogen role [38,39,40]. For example, *TNF-α* and *draper* were highly expressed in SF and SV, respectively. *IKK* and *IAP-1* were highly expressed in SV and CI compared with those in SF and CK, which may be due to the fact that these two genes play key anti-viral roles [39,40]. SF had higher expression of *SP7*, *and* lower expression of *IAP-1* than the other three treatments, which may be attributed to the anti-fungal role of these genes [13]. In addition, 218 (37.91%) DEGs were solely expressed in CI vs. CK, indicating that the co-invasion of RRSV and *M. anisopliae* YTTR may lead to these genes solely expressed in CI. *TNF-α* in SF was the highest expressed and the lowest in CI. We speculated that these two genes have an anti-fungal role and that the co-infection of RRSV and *M. anisopliae* YTTR suppressed the expression of *TNF-α*. The analysis of the KEGG pathways that enriched them in the top 20 pathways showed that DEGs were mainly enriched in immune-related pathways in the six comparison groups. It indicated that the pathogens’ invasion significantly initiated immune-related gene responses. In addition to the response of immune genes, the energy metabolic and synthetic pathways were also enriched, which means that the immune responses were energy-consuming processes requiring high energy levels to compensate for the consumption of immune responses [42,43,44]. These findings showed that there is a complex interaction between the insects and these pathogens. In this study, we mainly focus on humoral immune response, therefore, to further explore immune-related genes’ expression patterns in RNA-seq, 66 genes including in six major insect humoral pathways were selected. We found that, although some selected genes’ expressions were not significantly different among the four treatments, especially the Toll pathway genes, most of the genes had varying expression patterns in different treatments, or even showed completely opposite expression patterns between different treatments, such as *SP7*, *TNF-α, galectin-3, IKK,* and *IAP-1* [23,45,46,47]. At the same time, the expression of related genes in CI did not show the same expression pattern as that observed in SF, such as *IAP-1*, *lysozyme3*, *galectin-3*, and *TNF-α*, which might be due to the presence of RRSV, and thus these genes in response changed to different degrees [40]. The expression of the four genes (*Toll*, *Myd88*, *PPO*, and *TNF-α*) in our study have the same significance level or similar expression trend in three comparision groups compared with RNA-seq. Further, the expression level of the selected 19 genes that were used to validate transcriptomic data in four treatments using RT-qPCR was consistent with its expression level in RNA-seq, which proved the reliability of the RNA-seq approach. The results of RNA-seq and our study suggested that there is a complex interaction between both pathogens and insects. BPH showed different degrees of responses to the invasions of the different pathogens, while the presence of RRSV could induce more intricate immune responses of BPH after *M. anisopliae* YTTR infection. 

The largest number of selected genes were involved in the serine protease cascade. Additionally, this cascade plays an indispensable role in insect immunity by regulating antibacterial peptide synthesis, PPO activation, and melanin production [48,49,50]. It is also engaged in some vital physiological processes [51,52]. Hence, further study should mainly focus on the cascade, by exploring the mechanism of RRSV-meditated BPH to *M. anisopliae* infection. 

## 4. Conclusions

In summary, this study showed that *M. anisopliae* YTTR has a high virulence against BPH, and the presence of RRSV did not affect the susceptiblity of BPH to *M. anisopliae* YTTR infection. Transcriptome sequencing was used to explore the immune responses of RRSV-mediated BPH in response to *M. anisopliae* YTTR infection, where we established that immune-related genes expressed different responses to defend against different types of pathogens invasion. More genes were induced after the treatment of *M. anisopliae* YTTR when the RRSV was present in the host insect, while the presence of RRSV could also induce a more intricate immune response of BPH after *M. anisopliae* YTTR infection. In addition to the immune response, the expression of genes involved in crucial physiological processes was also significantly changed after pathogens invasion, especially for encountering RRSV and *M. anisopliae* YTTR infections. The findings of this study laid a basis to further clarify the immune response mechanism of RRSV-mediated BPH to *M. anisopliae* infection, and consequently, explore the immune regulatory mechanism of BPH in both pathogen infestations, digging more into resistance genes to improve the level of control of BPH in rice cropping systems. 

## 5. Materials and Methods

### 5.1. Materials

The test rice variety used was Taichung No. 1 (TN1), an insect and disease susceptible strain. RRSV-infected rice was tested by outer symptoms and RT-PCR, and positive samples were selected as RRSV-carrying rice for subsequent experiments [53]. 

BPH were long-term reared in an artificial climate chamber with 26 ± 1 °C, L:D = 14:10 and RH maintained at 70% ± 10%. They were continuously reared on TN1 for at least 10 generations prior to the bioassays. Newly molted BPH nymphs were used for virus acquisition by feeding them on RRSV-carrying rice. After two days of feeding, these nymphs were removed and placed on RRSV-free rice and waited for them to grow up into newly eclosion adults. RRSV-carrying BPH was also tested by RT-PCR. 

The test EPF used in the study was *M. anisopliae* strain YTTR, kindly provided by Dr. Shouping Cai, Fujian academy of forestry. The *M. anisopliae* strain YTTR was cultured on potato dextrose agar (PDA) medium and placed in an artificial incubator at 26 ± 1 °C and under dark conditions. 

### 5.2. RRSV Meditated the Virulence of Metarhizium anisopliae YTTR against BPH 

*M. anisopliae* YTTR conidia were harvested from 7 days culture by scraping the conidia into sterile centrifuge tubes on a sterile operating surface under a safe cabinet, and where sterile water containing 0.1% Tween-80 (named Tween-80) was added and vortexed for about 5 min to produce homogenous conidial suspensions. To determine the concentration of mother suspensions, conidial counts were made using a Neubauer Hemacytometer [54]. The conidial suspension was adjusted to appropriate working concentrations through serial dilutions prior to bioassays. Three concentrations (1.0 × 10^6^, 1.0 × 10^7^, 1.0 × 10^8^ spores/mL) were prepared using hemocytometer. The conidial suspension of each concentration was used to treat newly emerged adult BPH females (RRSV-carrying and RRSV-free) using a hand-held sprayer, while Tween-80 was used as a control treatment. Hence, four treatments, CK (Tween-80 against RRSV-free BPH), SF (single fungus infection, *M. anisopliae* YTTR against RRSV-free BPH), SV (single virus infection, Tween-80 against RRSV-carrying BPH), and CI (co-infection, *M. anisopliae* YTTR against RRSV-carrying BPH) were defined. Each treatment used a volume of 2 mL of suspension or Tween-80 to treat the insect. After air-drying, BPH were moved to a self-made BPH rearing device where rice seedlings were provided to them as a food source and changed every three days. The cultural solution, the Kimura B culture solution, was also changed every two days. The experiment was set up in Randomized Complete Block Design (RCBD) with three biological replicates where 30 BPH adult females were used per treatment. Mortality was observed daily, and the number of dead insects was recorded and then removed. 

### 5.3. Transcriptomic Sequencing

A previous study explored the RRSV-mediated effect on the expressions of immune genes of BPH to *M. anisopliae* YTTR infection at the concentration of 1 × 10^8^ spores/mL (Appendix A). RRSV-free BPH were treated with Tween-80 (CK) or *M. anisopliae* YTTR (SF), and RRSV-carrying BPH were also treated with Tween-80 (SV) or *M. anisopliae* YTTR (CI), respectively. The three insect major humoral pathways (Toll pathway, Prophenoloxidase (PPO) cascade, and c-Jun N-terminal kinase (JNK) pathway) were selected and the four major genes in these pathways were measured (*Toll* and *Myd88* in Toll pathway, *PPO* in PPO cascade and *TNF-α* in JNK pathway). It is commonly regarded that the Toll pathway was activated by G+ bacteria and fungi, however, a few studies found that the Toll pathway could be initiated by plant viruses [39,40]. JNK pathway and PPO cascade also could be activated after plant virus infection [36,38]. After 4, 8, 12, 24, 48, and 72 h post-treatment, selected genes’ expression were detected by RT-qPCR. Results showed that *M. anisopliae* YTTR and RRSV could induce different immune genes’ responses (Appendix A). In addition, the expressions of immune genes did differ between the single *M. anisopliae* YTTR treatment (SF) and the co-infection with *M. anisopliae* YTTR and RRSV (CI). Additionally, they even showed completely opposite expression patterns between SF and CI. Further, the expressions of immune genes were higher at the early stage (4–12 h post-treatment) of *M. anisopliae* YTTR infection than those at the later stage (24–72 h post-treatment). Hence, the time node of 12 h post-treatment was selected as the RNA sampling time for transcriptomic sequencing.

Samples from the four treatment groups (CK, SF, SV, and CI) as described above were collected 12 h after treatment. Three biological replicates were set up with 30 BPH adult females per treatment. The total RNA were extracted from the survived insects from the four treatments using the Trizol method. Additionally, the quality of the RNA was checked by Nanodrop 2000 (Thermofisher, Waltham, MA, USA) and agarose gel. The quality-checked RNA was enriched through oligo (dT), fragmented, reversed to cDNA, adding connectors, and then sequenced by Illumine platform, and these procedures were performed by Majorbio Biomedical Technology Co. (Shanghai, China). 

### 5.4. Transcriptomic Data Analysis

The raw data were assessed through quality control, removing adaptor, sequences with high N (N for uncertain base information) rates, and short-length reads. The data/reads (named clean reads) were then subjected to subsequent analysis. Clean data/reads were compared with the reference genome (https://www.ncbi.nlm.nih.gov/genome/?term=Nilaparvatalugens) (Access date: 24 September 2014) by using HISAT2 (http://ccb.jhu.edu/software/hisat2/index.shtml). Based on the comparison results, the clean reads were assembled and spliced into transcripts using StringTie (http://ccb.jhu.edu/software/stringtie/) and then annotated for analysis. Alignment of annotated genes was done with six major databases (NR (ftp://ftp.ncbi.nlm.nih.gov/blast/db/), Swiss-Prot (http://web.expasy.org/docs/swiss-prot_guideline.html), Pfam (http://pfam.xfam.org/), EggNOG (Clusters of Orthologous Groups of proteins, http://www.ncbi.nlm.nih.gov/COG/), GO (Gene Ontology, http://www.geneontology.org) and KEGG (Kyoto Encyclopedia of Genes and Genomes, http://www.genome.jp/kegg/)). Gene expression levels were compared by quantitative analysis using RSEM (http://deweylab.github.io/RSEM/) based on the value of FPKM (Fragments Per Kilobases per Million reads) (FPKM value > 0). To clarify the inner responses of BPH at the different treatment groups (CK, SF, SV, and CI), any two treatment groups were compared based on the DEGs analysis, and the six comparison groups (CI vs. SF, CI vs. CK, CI vs. SV, SV vs. CK, SF vs. CK, and SF vs. SV) were defined. The differential expressed genes (DEGs) were analyzed by Degseq 2 (ǀlog_2_FCǀ ≥ 1, *p* ≤ 0.05). Venn’s diagram was constructed to analyze the common or specific DEGs in different treatment comparisons. DEGs in each two-pair comparison were analyzed for KEGG enrichment, and these genes were enriched in the top 20 KEGG pathways using R scripts and Fisher’s exact test. Immune-related genes’ expressions were extracted from the transcriptome data and the relative expression levels of those genes compared with its expression in CK were analyzed. The data were analyzed on the online platform of Majorbio Cloud Platform (www.majorbio.com) [55].

### 5.5. Transcriptome Data Validation 

Randomly selected DEGs from transcriptome data were performed to verify the reliability of transcriptome data. RT-qPCR was used to verify the expression level of DEGs in BPH. According to the sequences of transcriptome data, special primers were designed, and β-actin was used as an internal reference gene (Appendix A) [56]. Total RNA was extracted from the four treatments (CK, SF, SV, and CI) using Trizol method as described above. The RNA was reverse transcribed for cDNA using a reverse transcription kit (Vazyme, Nanjing, China). Using the cDNA as a template, the expression level of each gene was detected using RT-qPCR. The protocols were conducted with reference to the kit (Vazyme, Nanjing, China). 

The expressions of the four genes that were previously measured (Appendix A) were used to detect the validation of transcriptome data. The four genes’ relative expression levels were computed by comparing their expression in treatments (SF/SV/CI) with CK. In addition, the relative expression levels of those genes between transcriptome data and in a previous study (Appendix A) were compared to further validate the reliability of transcriptome data. 

### 5.6. Data Analysis

All the data were analyzed using SPSS 22 and GraphPad Prism 9.0.0. The median lethal time (LT_50_) of *M. anisopliae* YTTR against the rice planthoppers (RRSV-free or RRSV-carrying) was analyzed using Probit analysis [57]. LT_50_ value means the number of post-treatment days for 50% of the test individuals that die, and a t-test was performed to analyze the significance level using SPSS 22. The survival rate of BPH was tested with a Logrank test (GraphPad) with significant differences at *p <* 0.05, 0.011, 0.001, and 0.0001. The relative expressions in the four treatments were counted by 2^−∆∆Ct^ [58] and analyzed using ANOVA followed by Duncan’s test (SPSS 22). The significance level of all tests was set at *p* < 0.05 or *p* < 0.01. The data of RNA-seq were analyzed using the online platform of Majorbio Cloud Platform (http://www.majorbio.com) [55]. 

## Figures and Tables

**Figure 1 plants-12-00345-f001:**
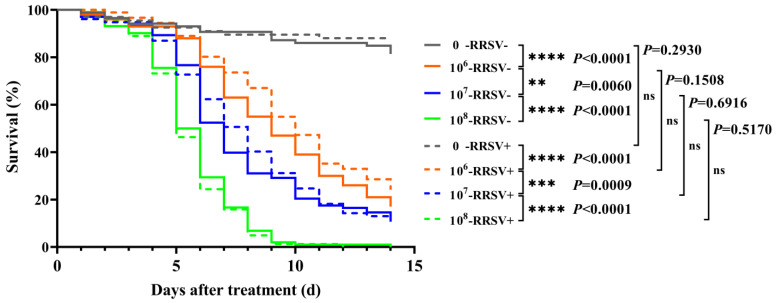
Survival curves of BPH showing RRSV-meditated lethal effects of *Metarhizium anisopliae* YTTR against BPH. Data are mean ± SEM (n = 30), and significant differences were represented by definite *P* value and asterisk (** *p <* 0.01; *** *p <* 0.001; **** *p <* 0.0001) (Logrank test). In fact, 0, 10^6^, 10^7^, and 10^8^ are the various concentrations of *M. anisopliae* YTTR at 0, 1 × 10^6^, 1 × 10^7^, and 1 × 10^8^ spores/mL, respectively. RRSV– and RRSV+ represent RRSV-free or RRSV-carrying BPH respectively.

**Figure 2 plants-12-00345-f002:**
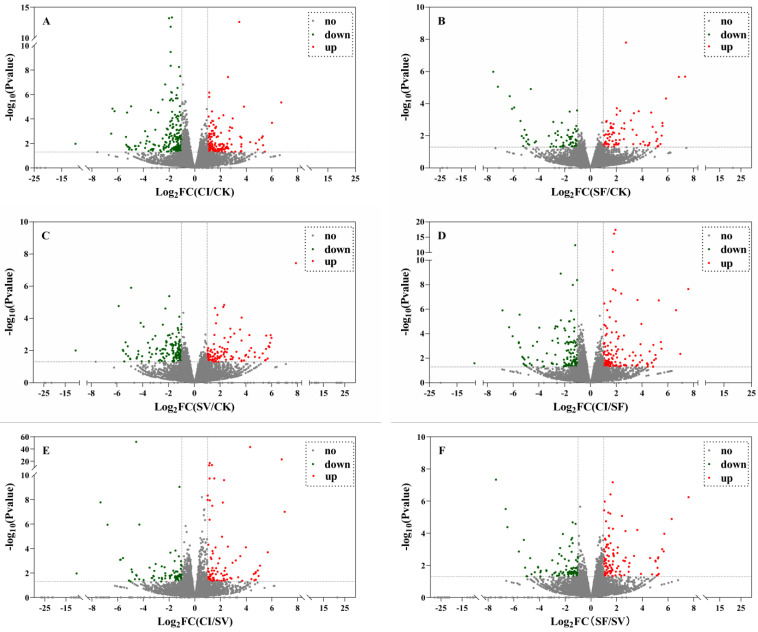
Volcanic plots of DEGs in each combination group. RRSV-free BPH were treated with Tween-80 (CK) or *Metarhizium anisopliae* YTTR (SF) and RRSV-carrying BPH were also treated with Tween-80 (SV) or *M. anisopliae* YTTR (CI), respectively. After 12 h, RNA-seq was performed. The expression of DEGs among the four treatment groups (CK, SF, SV, and CI) were compared with any two treatment groups. CI vs. CK, SF vs. CK, SV vs. CK, CI vs. SF, CI vs. SV, and SF vs. SV were defined. (**A**–**F**) respectively means the comparison group of CI vs. CK, SF vs. CK, SV vs. CK, CI vs. SF, CI vs. SV, and SF vs. SV.

**Figure 3 plants-12-00345-f003:**
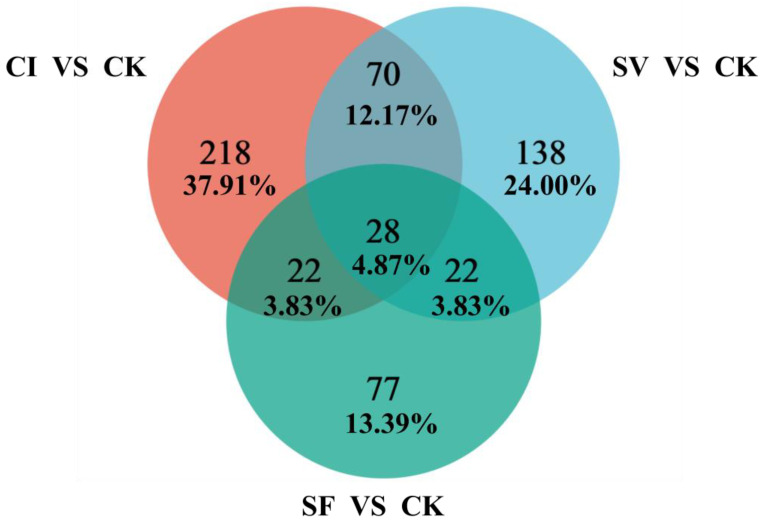
Venn’s diagram of DEGs in comparison groups. RRSV-free BPH were treated with Tween-80 (CK) or *Metarhizium anisopliae* YTTR (SF) and RRSV-carrying BPH were also treated with Tween-80 (SV) or *M. anisopliae* YTTR (CI), respectively. After 12 h treatment, RNA-seq was performed. The expression of DEGs among the four treatment groups (CK, SF, SV, and CI) were compared with any two comparison groups. CI vs. CK, SV vs. CK, and SF vs. CK were defined.

**Figure 4 plants-12-00345-f004:**
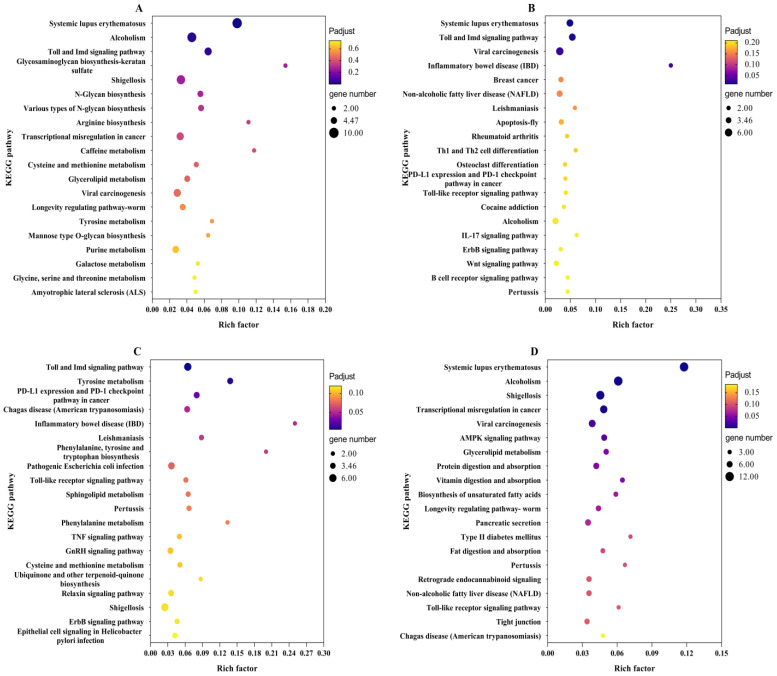
Enrichment map of DEGs in KEGG pathways. RRSV-free BPH were treated with Tween-80 (CK) or *Metarhizium anisopliae* YTTR (SF) and RRSV-carrying BPH were also treated with Tween-80 (SV) or *M. anisopliae* YTTR (CI), respectively. After 12 h treatment, RNA-seq was performed. The expression of DEGs among the four treatment groups (CK, SF, SV, and CI) were compared with any two comparision groups. CI vs. CK, SF vs. CK, SV vs. CK, CI vs. SF, CI vs. SV, and SF vs. SV were defined. (**A**–**F**) respectively means the result of the enrichment map in the comparison group of CI vs. CK, SF vs. CK, SV vs. CK, CI vs. SF, CI vs. SV, and SF vs. SV.

**Figure 5 plants-12-00345-f005:**
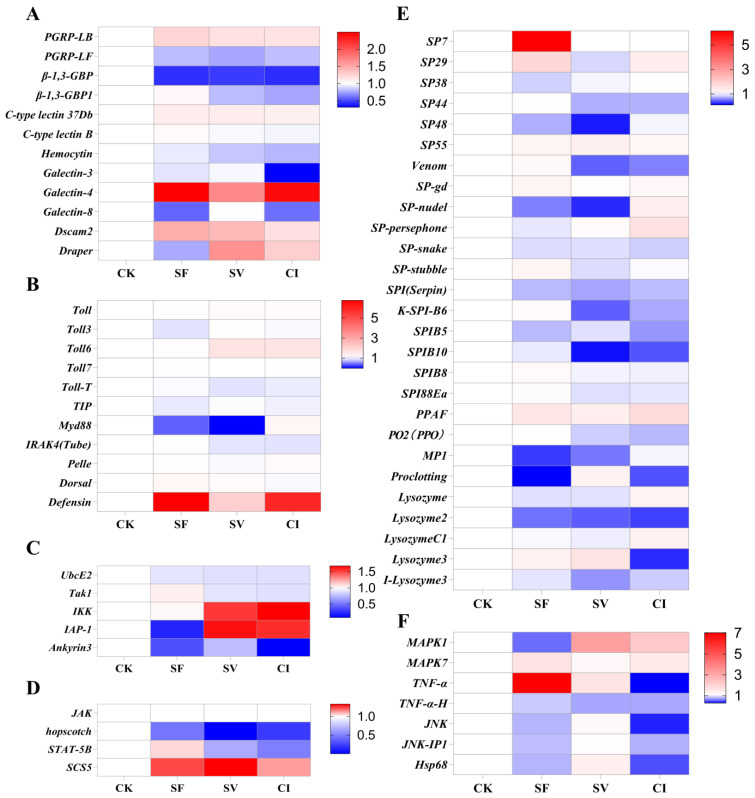
Relative expression levels of immune-related genes in RNA-seq. RRSV-free BPH were treated with Tween-80 (CK) or *Metarhizium anisopliae* YTTR (SF), and RRSV-carrying BPH were also treated with Tween-80 (SV) or *M. anisopliae* YTTR (CI). (**A**–**F**) respectively means the pathogen recognition receptors (PRRs), Toll pathway, immune deficiency pathway (IMD pathway), Janus kinase/signal transducers and activators of transcription pathway (JAK-STAT pathway), Serine protease cascade and Prophenoloxidase cascade (PPO cascade) and c-Jun N-terminal kinase pathway (JNK pathaway). The heatmap was constructed based on the relative expressions of immune genes compared to CK in RNA-seq. Different colors mean different multiples, and red indicates high expression and blue indicates low expression.

**Figure 6 plants-12-00345-f006:**
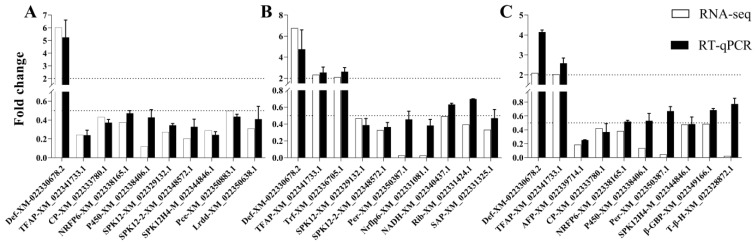
Transcriptomic data validation by RT-qPCR. RRSV-free BPHs were treated with Tween-80 (CK) or *Metarhizium anisopliae* YTTR (SF), and RRSV-carrying BPHs were also treated with Tween-80 (SV) or *M. anisopliae* YTTR (CI). (**A**–**C**) respectively means the expressions of DEGs among the three comparision groups of CI vs. CK, SF vs. CK, and SV vs. CK.

## Data Availability

The data presented in this study are available on request from the corresponding author.

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
