# Peer review of "Immune Responses and Transcriptomic Analysis of Nilaparvata lugens against Metarhizium anisopliae YTTR Mediated by Rice Ragged Stunt Virus"

_plants, 2023, doi:10.3390/plants12020345_

Round 1
Reviewer 1 Report (Previous Reviewer 2)
I base this review on the letter submitted by the authors addressing my comments. It appears the authors have addressed my comments, so I recommend publication.
In this review I was unable to evaluate the pdf of the revised manuscript. There are so many track changes that it is difficult to figure out what to read and which figures and figure legends are applicable.
Author Response
Dear reviewer,
Please see the attachment.

Reviewer 2 Report (New Reviewer)
Li et al used a virulent strain of EPF (Metarhizium anisopliae YTTR) to explore the RRSV-mediated immune response of BPH using RNA-Seq technology. Their findings indicated that M. anisopliae YTTR showed high virulence against BPH. Moreover, the presence of RRSV did not affect the susceptibility of BPH to M. anisopliae YTTR infection. This interesting study might provide essential information to the researchers working in this area. However, I would like to suggest further improving the quality of the manuscript by adding explanatory methodology and logical discussion to improve the quality of the manuscript. Moreover, several typos and grammatical issues also need to be corrected thoroughly.
Results
Line 133-135: Truncated information. Please complete it.
Line 144: Suggestion. Is it possible to explain the results of the Venn diagram in percentage so that the reader could easily know how much percent genes were induced in a particular treatment or overall treatments?
Discussion
Line 283-299: This part seems just a repetition of the results section—no logical discussion.
The discussion section should be developed with more logical details about the change in expression patterns of the genes in different treatments. Moreover, In the whole manuscript, I was expecting that there would be a strong focus on antiviral and antifungal immune genes and how they behaved as the treatments included viruses and fungus; however, I was unable to find such information. Moreover, the authors mentioned that they made a random selection of genes for qPCR validation, which is also questionable when you are focusing on immune genes, especially those related to viruses and fungi having significant differential expression or abundant expression should have been selected.
Methods
Line 349: BPH or N. lugens? Please be consistent with one name in the whole manuscript.
Line 352: What do you mean by needed stage? Please clearly describe which stage was used for further experiments.
Line 416: What were the criteria for selecting contigs or genes based on the value of FPKM? There is no information here
Line 428: Transcriptome data validation
actin was used as an internal control; I am not sure whether the authors validated it by themselves or previous studies validated it in this particular situation, i.e., after pathogenic treatment. In both cases, it should be mentioned clearly, as the selection of an unvalidated reference gene might lead to erroneous results.
Line 438: What was the reason behind selecting these four genes (Toll, Myd88, PPO, and TNF-α) in pilot study selection and for further validation? Please explain clearly. Moreover, figure S2 should be modified, and the genes' names should be mentioned instead of A,B, etc. In this way, the figure might become self-explanatory.
Round 2
Reviewer 2 Report (New Reviewer)
The authors have addressed the comments, and the manuscript can be accepted now.
This manuscript is a resubmission of an earlier submission. The following is a list of the peer review reports and author responses from that submission.
Round 1
Reviewer 1 Report
The manuscript entitled: ‘Immune response and transcriptomic analysis of rice virus-mediated response to Metarhizium anisopliae infection in Nilaparvata lugens’ aimed analyse the M. anisopliae impact on BPH carrying rice ragged stunt virus (RRSV) or not by the determination of relevant induced mortality parameters, as well as, evaluate BPH response to RRSV and M. anisopliae by means of RT-qPCR and RNA-seq. The manuscript raise an interesting topic, however, I find this work is not suitable for the publication in the present form.
The main objection is the way the results were presented. The authors performed RT-qPCR analysis followed by RNAseq. These analyses were performed here independently, while, they can be easily confronted with each other. The consistency of the results obtained in RT-qPCR and transcriptome analyses was only mentioned in the Discussion section without giving any examples. Authors examined the expression of the chosen immune genes by means of RT-qPCR. What was the level of their expression in RNAseq data? Were these among the 77 immune-related DEGs? It is hard to find it as the accession numbers of the chosen genes for RT-qPCR (XM_0223…) are different from of those 77 DEGs (gene-LOC111…). A table presenting the expression values in each treatment as well as the description of those 77 DEGs would be beneficial.
Moreover, authors stated in the main text that 12 samples were mapped (line 218) in the RNAseq analysis, while in the Table S3 it is presented that 3 samples were analysed for each treatment (CK, SF, SV) except for CI, where there are only 2, which gives 11 samples in summary. For such a low number of samples per treatment, the resulting data cannot be reliable. Moreover, no DEGs responsible for specific interactions were presented. The RNAseq data is presented very generally.
Title:
I don’t find the title is delineated grammatically. Moreover, the phrase ‘rice virus’ is too general. Here, only one virus was studied, therefore the whole pathogen name can be specified.
Results:
A Venn’s diagram could be applied in transcriptomic analysis to show if there are DEGs that are common or not for the performed treatments comparisons. Here, DEGs that may be responsible for those interactions may be pinpointed.
Other comments:
44, 53, 75, 303, 310, 334, 351, 468: Authors should use one word.
59-60: The full name Metarhizium anisopliae was used above.
109: LT50 – explain what stands for this parameter
146-164: explain the abbreviations for CI, SF, CK, SV
146-164: what proteins are encoded by those immune genes, PGRP-LB and Myd88
146-212: A phrase ‘was significantly higher’ is too frequently used.
223: inconsistency in figure numbering
393-394: This was already stated in lines 389-390
408: ‘America’ is not a country, as well as ‘Thermo’ is not a full name of the producer
423-427: How the material subjected to RNAseq was prepared? I haven’t found an information concerning the sample preparation for RNAseq. What time after treatment the insects were collected? Was the quality of the RNA subjected to RNAseq checked (RIN)?
449: The verification of the results obtained from RNAseq is called validation.
486: There are no contribution for Minsheng You and Sheng Lin.
497-498: Raw data from RNAseq should be deposited in a repository
Figures:
Bad quality of figure labels. Very small axis labels make impossible to read. Moreover, the graph lines in the Figure 2A are almost indistinguishable.
Figures 4-7: there are no figure explanations.
Figure 4. These should be named ‘Volcano plots’.
Author Response
Dear reviewer,
We sincerely thank you for your constructive comments concerning our manuscript (Manuscript ID: plants-2002648). These professional and detailed comments are all valuable and helpful for improving our article quality. We have revised them one by one according to your suggestions and tried our best to modify our manuscript. Please see the attachment.

Reviewer 2 Report
This article on a plant hopper investigates whether infection with rice ragged stunt virus affects the lethality of Metarhizium anisopliae, and whether co-infection also affects gene expression. The authors conclude that co-infection does not increase lethality, but that it induces the expression of more immune-related and metabolic-related genes.
This is an interesting study with a significant amount of research. However, significant flaws in the presentation of the data prevent the reader from understanding the effects of co-infection, especially as pertains to gene expression.
Major points:
1. The manuscript uses too many abbreviations, which makes it difficult to read. For example, the control “ck” is poorly defined, and some sentences are very difficult to follow (for example, “The four treatment groups were compared two by two (CI vs SF, CI vs CK, CI vs SV, SV vs CK, SF vs CK, and SF vs SV)”). Clarity should be increased.
2. Survival experiments. The statistics used to analyze survival are unclear. First, the entire survival curves should be analyzed using the Logrank test, which GraphPad (used by the authors) can do. Second, the LT50 and cumulative mortality were analyzed by one-way ANOVA followed by Tukey’s test, but the data have two variables: concentration and RRSV status. Therefore, they should be analyzed by two-way ANOVA, followed by a post-hoc test (like Tukey’s).
3. There is a significant message problem in section 2.3. First, the authors should conceptualize why they examined the Toll, PO and JNK pathways specifically, and excluded pathways like IMD and Jak-Stat, especially because the IMD pathway was a hit in the RNAseq experiment. A rational explanation would suffice. Second, rather than presenting the findings so specifically by gene, the authors should present the results in terms of (1) what they say about the pathways (not just the individual components) and (2) what they say about the effect of infection or co-infection. As written, the message as pertains to the main question of the study is unclear or lost.
4. In Figure 3, spell out the treatments.
5. There is also a significant message problem in section 2.4. What does the data tell us about the main story? Are there specific immune or metabolic pathways that stand out?
6. In figure 5, some of the Kegg pathways that are highlighted in the panels are uninformative. It seems like the output of the analysis was placed as a figure without looking at it with a conceptual lens. For example, what does systemic lupus and alcoholism tell us about what is happening in these plant hoppers?
7. Figure 6 is not particularly informative because the genes are unknown to the reader. Are there pathways that are overrepresented and inform on the main question?
8. PCR validation of RNAseq data. Generally, I do not believe that PCR validation is necessary, but it is a good addition. However, the reader will be interested to know whether the RNAseq data agrees with the RT-PCR experiments on immune genes that are presented in figure 3.
9. The RNAseq data needs to be deposited in a public repository (one option is NCBI), and an accession number must be provided. This is standard practice.
Minor points:
1. Line 74: The sentence is about immune responses in general, yet the reference only covers fungi. Consider adding additional references.
2. Lines 79-80: In my opinion, omics techniques have led to an upsurge in studies on insect immunity in general, and not just the humoral immune response.
3. Line 123: Remove the parentheses from the 2.2 section title to differentiate the section from 2.1.
4. Line 223: I believe the authors mean Figure S2A (and not Figure 10).
5. The conclusion section should go after the discussion section and not after the methods.
6. This study does not have subjects, so the “informed consent” section should be deleted.
Author Response

(The authors gave the same response as above.)

Round 2
Reviewer 1 Report
The authors responded to all questions, however, I still have concerns associated with the presented data. My biggest concern is the inconsistency of the RT-qPCR results with RNAseq data, which was explained by the authors by difference in virulence. In my opinion, despite those differences, the trend should be concordant. If those immune genes analysed in RT-qPCR are so relevant in the Nilaparvata lugens-Metarhizium anisopliae-RRSV interactions, this should be supported by RNAseq data. Especially when authors performed RT-qPCR analysis over time course to select specific time after treatment to do RNAseq.
Other comments:
The figures still have bad quality. In some cases - even worse.
Figure 3. The information concerning statistically significant results should be added to this figure.
In my opinion, KEGG terms associated with human diseases should not be presented in the main text. Is there any connection between human diseases and insect response to fungi/virus?
Figure 8. Were the results of RT-qPCR done in order to validate transcriptomic data statistically significant?
Author Response

(The authors gave the same response as above.)

Reviewer 2 Report
This revision is an improvement over the original submission, but I still have significant qualms regarding the study. In fact, the author response to one of my comments makes me call into question the practical validity of some of the key data.
Major concern: In my review of the original submission, I asked whether there was concordance between the qPCR data in the pilot study and the RNAseq data in the later study. According to the authors, the two datasets are not in agreement with one another. The authors postulate that this could be due to the intensity of infection being different between the two experiments (the infection intensity is unknown). If that were the case, I would still have expected concordance, but with a difference in magnitude. However, this does not appear to be the case. And even if the author contention that different infection intensities can yield different transcriptomes is true, the fact that two very different transcriptomic profiles arise from the same infection means that the data presented in this study is only applicable to the experiment in this study and cannot be used to predict what would happen in similar experiments or in nature. In other words, the data do not teach us about the biology of the interaction, and hence, has limited value.
Other comments:
1. I am still confused as to the practical value of mapping DEGs to human diseases such as alcoholism, shigellosis, inflammatory bowel disease, etc. What does that teach us about insects? I do not question that this is the output from the analysis, but it does not seem relevant to insects.
2. In line 309, the authors write that the Toll and IMD pathways were enriched but may play a non-special role against pathogens. Why? What is the rationale behind this argument?
3. In line 500 the authors write that “the presence of RRSV significantly enhanced the susceptibility [sic] of BPH against M. anisopliae infection”. I do not believe this is supported because figure 2 shows that RSSV infection does not alter insect survival.
4. My evaluation written above is based on the science. If the journal decides to pursue this submission further, then the manuscript will require significant editing for language and clarity.
Author Response

(The authors gave the same response as above.)
